# Post-Operative Results of ACL Reconstruction Techniques on Single-Leg Hop Tests in Athletes: Hamstring Autograft vs. Hamstring Grafts Fixed Using Adjustable Cortical Suspension in Both the Femur and Tibia

**DOI:** 10.3390/medicina58030435

**Published:** 2022-03-16

**Authors:** Lokman Kehribar, Ali Kerim Yılmaz, Emre Karaduman, Menderes Kabadayı, Özgür Bostancı, Serkan Sürücü, Mahmud Aydın, Mahir Mahiroğulları

**Affiliations:** 1Department of Orthopaedics and Traumatology, Samsun University, Samsun 55090, Turkey; lokmankehribar@gmail.com; 2Faculty of Yaşar Doğu Sport Sciences, Ondokuz Mayıs University, Samsun 55270, Turkey; emre.karaduman@omu.edu.tr (E.K.); menderes@omu.edu.tr (M.K.); bostanci@omu.edu.tr (Ö.B.); 3Department of Orthopaedics, University of Missouri, Kansas City, MO 64108, USA; serkansurucu@outlook.com; 4Haseki Training and Research Hospital, Orthopaedics and Traumatology, Istanbul 34096, Turkey; mahmut_aydn@windowslive.com; 5Memorial Sisli Hospital, Orthopaedics and Traumatology, Istanbul 34384, Turkey; mahir.mahirogullari@memorial.com

**Keywords:** anterior cruciate ligament reconstruction, cortical suspensory fixation, interference screw, single-leg hop test, return to sport

## Abstract

*Background and Objectives*: Anterior cruciate ligament (ACL) tears are common injuries in the athletic population, and accordingly, ACL reconstruction (ACLR) is among the most common orthopedic surgical procedures performed in sports medicine. This study aims to compare the semitendinosus/gracilis (ST/G) and ACL hamstring grafts fixed using adjustable cortical suspension in both the femur and tibia (MAI) ACLR techniques. We aimed to compare the results of single-leg hop tests (SLHT) applied in different directions and limb symmetry indices (LSI) in athletes with a 6-month post-operative ACLR history. *Materials and Methods*: A retrospective cohort of 39 athletes from various sports branches who underwent MAI (*n* = 16) and ST/G (*n* = 23) ACLR techniques by the same surgeon were evaluated. The knee strength of the participants on the operated and non-operated sides was evaluated with five different SLHTs. The SLHT included the single hop for distance (SH), triple hop for distance (TH), crossover triple hop for distance (CH), medial side triple hop for distance (MSTH), and medial rotation (90°) hop for distance (MRH). *Results*: There was a significant improvement in the mean Lysholm, Tegner, and IKDC scores in the post-operative leg for both techniques (*p* < 0.05) compared to the pre-operative levels. When there was a difference between the SH of the operative and the non-operative legs in the ST/G technique (*p* < 0.05), there was no significant difference in the other hop distance for both ST/G and MAI (*p* > 0.05). There was no difference between the techniques regarding the LSI scores. *Conclusions*: The fact that our research revealed similar LSI rates of the SLHTs applied in different directions in the ST/G and MAI techniques assumes that the MAI technique can be an ACLR technique which can be functionally used in athletes.

## 1. Introduction

Anterior cruciate ligament (ACL) tears are common injuries in the athletic population, and accordingly, ACL reconstruction (ACLR) is among the most common orthopedic surgical procedures performed in sports medicine [1,2,3]. Today, the quadriceps tendon, patellar tendon, and semitendinosus/gracilis (ST/G) tendons are the most commonly used graft types in ACLR [4]. In addition, a new technique called “all-inside” using four-strand (4ST) grafting with only the ST tendon has become widespread [5]. Another important aspect for ACLR is the graft-fixation method. Fixation to the femoral cortex proximally provides good results, while tibial fixation is usually performed with an interference screw [6,7]. The all-inside technique with a distal cortical fixation in the femur and tibia is a new technique that was defined by Lubowitz et al. [8] in 2011. Thanks to this technique, the retrograde drilling of shorter tibial tunnels is achieved, thereby limiting bone loss, which, in turn, enables graft stabilization with an adjustable fixation system [9]. Researchers reported that this fixation method allows for more active rehabilitation in patients [10]. In spite of the advantages of the all-inside technique, the need to use a special drill while creating the socket, the problem of adjusting the socket depth, and its high cost [11,12] led physicians to develop the modified ACL surgery technique (MAI) [13]. In the MAI technique, the ST graft, which is prepared by stranding the ST tendon four times, is fixed on the tibia and femur with suspensory fixation.

Single-leg hop tests (SLHT), which are frequently used during rehabilitation processes after ACLR and in decision making for returning to sports (RTS), are commonly performed to evaluate the functional status of athletes, to reveal the limb asymmetries between the operated and non-operated sides, and to follow the developments in the operated limb [14,15,16]. The biggest limitation of the conventionally applied SLHTs is that they mostly consist of straight forward movements [14,17], whereas during sports activities, movements are applied in multiple directions. Therefore, the SLHTs that are only applied straight forward can be insufficient in evaluating lower-extremity performance, in preventing injury, or in monitoring RTS after surgical operations such as ACLR [18,19]. Researchers have reported that multi-directional tests are of great importance in addition to the SLHTs that are performed straight forward, particularly when returning to sports after injury [14,20]. In fact, increased asymmetry rates have been reported in the multi-directional tests compared to the traditionally applied SLHTs [21]. Therefore, it can be argued that the multi-directional jump tests that are applied in addition to the traditional SLHTs after ACLR play a critical role in determining the rehabilitation and the time to RTS.

Based on all of this information, this study focused on two objectives. The first of these was to compare the SLHT results in athletes with a 6-month post-operative ACLR history to whom ST/G and MAI ACLR techniques had been applied, and the second was to evaluate the limb asymmetries that occurred in the SLHTs applied both in the forward direction and in the medial and rotational directions.

## 2. Materials and Methods

### 2.1. Participants

This study was approved by the Clinical Research Ethics Committee at the Samsun Training and Research Hospital (Approval No: GOKA/2021/17/14) and was conducted in accordance with the Declaration of Helsinki. A retrospective cohort of 39 athletes from various sports branches who underwent the MAI (*n* = 16) and conventional ACL reconstruction (*n* = 23) techniques by the same surgeon between April 2019 and March 2021 were evaluated. Patient records were analyzed to obtain the data.

The inclusion criteria for the study were as follows: patients aged 18–35 years, isolated ACL rupture in only one knee without concomitant meniscal, chondral, or other ligamentous injuries, no other neuromuscular or musculoskeletal injuries, and no history of contralateral knee surgery or injury. Sixth-month post-operative Lysholm, Tegner, and International Knee Documentation Committee (IKDC) [22,23,24] scores of the patients were evaluated. To reduce variability during the recovery period, all participants were referred to the same rehabilitation program following surgery.

The semitendinosus and gracilis tendon autografts from the same leg were used in the ST/G ACL reconstruction. Both tendons were doubled to form a four-stranded graft. A closed socket was drilled into the femur via the medial arthroscopic portal; an open tunnel was drilled into the tibia from the outside. Suspensory fixation was used to fix the graft to the femur, and interference-screw fixation was used to fix it to the tibia. In the MAI technique, only the semitendinosus tendon was harvested, and its preparation was quadrupled. The graft was fixed to the femur and tibia using a suspensory-fixation device.

### 2.2. Procedures

The Lysholm, Tegner, and IKDC scores (pre- and post-operative) and the 5 different SLHT (6 months post-operative) performances of all participants were determined. For these measurements, all participants visited the laboratory 3 times in total, except for the routine post-operative controls. In the first visit (pre-operative), the participants filled in subjective questionnaires consisting of Lysholm, Tegner, and IKDC scales and were informed about the study. In the second visit (6 months post-operative), the anthropometric data were obtained, and the participants were familiarized with the SLHTs planned for the next visit (familiarization). In the third laboratory visit (3 days after the second visit), the participants filled in the Lysholm, Tegner, and IKDC scales for the second time (post-operative), and the SLHT performances were measured.

The knee strength and functions of the participants on the operated and non-operated sides were evaluated, with the SLHT consisting of 5 different series. The SLHT included the single hop for distance (SH), triple hop for distance (TH), crossover triple hop for distance (CH), medial side triple hop for distance (MSTH), and medial rotation (90°) hop for distance (MRH) over a single line with maximum effort [25,26]. These tests demonstrated good test–retest reliability in patients after ACL reconstruction [26].

For the SLHT, the ground was marked with a 6 m long, 15 cm wide tape running perpendicular to the start and finish lines. For the SH, participants were instructed to stand on the leg to be tested, jump forward as far as possible, and land with the same leg. For the TH, participants were instructed to stand on the leg to be tested, perform three consecutive maximum forward jumps, and land with the same leg. For the CH, participants were instructed to stand on the leg to be tested then jump as far forward as possible 3 times in a row while crossing the 15 cm (width) marked strip. The hop distance was measured to the nearest centimeter from the starting line to the patient’s heel with a standard tape measure. For the MSTH, participants were instructed to stand on the leg to be tested with the medial side of the foot perpendicular to the direction of jump. The participants jumped forward as far as possible on the same leg three times in a row. The total distance of the three consecutive jumps was measured from the medial part of the foot in the starting position to the medial part of the foot at the time of the landing. For the MRH, participants were instructed to stand on the leg to be tested with the medial side of the foot perpendicular to the direction of the jump. A jump was performed in the transverse plane, and the participants were asked to rotate their foot 90° in the medial direction during the swing phase. Before takeoff, the participants were not allowed to turn their foot in the jump direction. Care was taken to ensure that the foot position was perpendicular to the jump direction (N10° deviation from the rolling meter). This position was visually checked by the supervisor. The hop distance was measured from the medial side of the foot at takeoff to the toes at landing (Figure 1). All jumps were considered successful if the landings were stable. The post-jump landing was approved when it was under the participant’s full control and on the tested limb. The test was repeated if the participant lost balance, touched the wall, or had additional bounces after landing.

### 2.3. Statistical Analysis

SPSS 22.0 for Windows (SPSS Inc., Chicago, IL, US) was used for statistical analyses, and a two-sided *p*-value < 0.05 was considered as statistically significant. Descriptive data were presented as the mean and standard deviation. The data were checked for normality using the Kolmogorov–Smirnov test and were examined for kurtosis and skewness. Comparisons between the groups over time were determined using a mixed repeated-measures analysis of variance (RM ANOVA) with a Bonferroni correction for post hoc analyses. The Independent *t*-test was also used to evaluate the differences in the limb symmetry index (LSI) between the techniques (MAI and ST/G). All graphics were performed with the GraphPad Prism software (version 9.2.0, GraphPad Software Inc., San Diego, CA, USA).

## 3. Results

A total of 39 male athletes were enrolled in the study. The cohort consisted of 16 MAI (41%) and 23 ST/G (59%) techniques. There was no statistically significant difference between the groups according to age, weight, height, BMI, and time to RTS (Table 1).

Compared to pre-operative levels, there was a significant improvement in the mean Lysholm, Tegner, and IKDC scores at the post-operative level for both techniques (for all, *p* < 0.05). In the ST/G group, the Lysholm scores were 91.7 ± 9 and 95.8 ± 8.4, the Tegner scores were 6.2 ± 1.2 and 5.7 ± 1.4, and 50.3 ± 8.3 and 89.2 ± 7.4 for the IKDC subjective scores, pre- and post-operatively, respectively. In the MAI group, the corresponding Lysholm scores were 74.5 ± 12.5 and 97.1 ± 4.2, for the Tegner were 6.7 ± 1.7 and 6.3 ± 1.9, and for the IKDC were 50.7 ± 9.4 and 92.2 ± 6.3. There were no differences between techniques with regard to the outcome measures, including the Lysholm, Tegner, and IKDC scores (*p* > 0.05) (Figure 2).

When there was a difference between the SH distance of the operative and non-operative legs in the ST/G technique, there was no significant difference in the other hop distances. In the MAI technique, all hop distances were not statistically different between the operative and non-operative legs. There were significant differences between the operative and non-operative legs in the SH (158.6 ± 13.7 vs. 164.7 ± 14.3) (*p* < 0.05), but there were no differences in the TH (550.7 ± 55 vs. 560.3 ± 52.9), CH (514.8 ± 38.5 vs. 514.4 ± 45.6), MSTH (457.1 ± 46.8 vs. 466.6 ± 50.1), and MRH (145 ± 12.7 vs. 147.7 ± 13.4) distances in the ST/G technique (*p* > 0.05). In the MAI technique, the corresponding results were 160.5 ± 17 vs. 165.9 ± 18 cm, 545.5 ± 64.6 vs. 554.6 ± 51.9 cm, 512.4 ± 59.5 vs. 516.4 ± 60.9 cm, 454.8 ± 53.7 vs. 460.4 ± 58.2 cm, and 142.9 ± 10.9 vs. 144.6 ± 13.6 cm, respectively (for all, *p* > 0.05) (Figure 3).

As shown in Table 2, there was no difference between techniques regarding the LSI scores (*p* > 0.05).

## 4. Discussion

When the main findings of our research were evaluated, the results were as follows: pre-operative and post-operative findings were similar in terms of the Tegner, Lysholm, and IKDC scores in both the ST/G and MAI ACL reconstruction techniques. While the TH, CH, MSTH, and MRH performances of the patients’ operated and non-operated sides were similar in both the ST/G and MAI techniques, a difference was only observed in the ST/G technique of the SH performance. Finally, the results of this study showed that the LSIs of both techniques were similar.

The LSI, which is calculated using the SLHTs, is a measure that can be easily used to determine the difference between the legs of healthy individuals, individuals with disabilities, or those with a history of lower-extremity operation. Studies conducted on healthy individuals and athletes without a disability generally found that the difference between the limbs was in the range of 10–15% at most in the four traditionally measured SLHTs (SH, TH, CH, and 6 m. timed hop test) [27,28].

The time to RTS and the ability to continue sports activity are two important outcomes after ACL reconstruction. The return to sport is a confirmation of full lower-limb functional recovery. Additionally, it can be a good strategy for full functional recovery and the completion of rehabilitation [29]. The LSI, which is one of the evaluation criteria for returning to sports following ACL reconstruction (ACLR), was evaluated by researchers after different ACL techniques were used, and different findings have emerged as a result of these evaluations. Barford et al. [30] measured the SH of the subjects at 6 and 12 months after ACLR was conducted with the hamstring autograft technique and found LSI rates of >85 in the majority of the group. Herrington et al. [31] evaluated the isokinetic and isometric quadriceps strengths and the SH and CH strengths of professional football players after ACLR and found that the LSI ratios did not reach >90, but the difference between the two limbs was not in the re-injury risk range. The researchers also reported that the results of the SLHTs did not overlap with the isokinetic and isometric strength results and that although the LSIs in the SLHTs were acceptable within the normal range, much lower strength rates in terms of the quadriceps strength of the subjects were found in the isokinetic and isometric tests compared to the healthy side. Our study showed that the 6-month post-operative LSI rates were within the range of <10% in athletes to whom both the ST/G and MAI ACLR techniques had been applied.

Although researchers have recently reported that SLHTs alone should not be an evaluation criterion for RTS in particular since they do not exactly reflect quadriceps strength after ACLR [31], some researchers have reported that at least two SLHTs that are applied in different directions might reveal important results for RTS [12,32]. In fact, in our study, five different SLHTs that were applied in different directions were evaluated, and similar findings were found in terms of the LSI, both statistically and in percentages. Dingenen et al. [12] emphasized that the asymmetric ratios found in their study might vary depending on the tests performed. These researchers stated that although statistical differences were not observed in the LSIs, there might be differences in clinical decision making depending on the threshold values. In fact, in their study, they found >90% similarity between the limbs in all of the subjects in the traditional SLHTs, while revealing >90% similarity in only 68.8% of the subjects in the medial and rotational jump tests. Similarly, in a study conducted not only on individuals with an ACLR history but also on healthy and athletic groups, no limb asymmetry was detected in the traditional SLHTs, while asymmetrical rates were reported in the MSTH [33]. These findings clearly indicate that not only the unidirectional SLHTs but also the medial and rotation hop tests should be used in decision making for RTS after ACLR.

Clinically, it is important to conduct tests that are applied jointly on different sides because a ≥90% LSI is usually for the SH and TH, and this ratio should be considered for RTS [18]; however, the medial and rotational tests to be applied alone may reveal low asymmetry rates, and this may prolong the RTS times. Therefore, the findings of the SLHT applied in different directions are of great importance in order to make the most objective evaluation of the operated sides, particularly after ACLR. A systematic review study also supports this result [18]. The fact that the medial and rotationally applied SLHTs revealed asymmetrical ratios compared to the traditionally applied ones in some studies is still an unclear and controversial result. However, researchers have pointed out that the biomechanical aspects of lower-extremity biomechanics might vary according to the direction of the jump and landing, as well as dynamic postural stability [34,35,36]. After the medial and rotational jump, the hip abduction, medial rotation, and knee valgus may restrict movement during the landing, and in this case, knee-related injuries may occur [26,37]. Therefore, positive results from medial and rotational tests that are applied after ACLR can give us important information about the health of the knee joint.

However, as in our study, for clinical decision making, it is important to evaluate training components such as balance, strength, quickness, and agility, which are affected by the lower extremities, in addition to the SLHTs applied in different directions in athletes with an ACLR history. This limited us to objectively evaluate the return of the athletes in our study to physical activity with maximum performance. In addition, one of the most important limitations of our study was that the tests were planned and were conducted in a closed environment without causing any fatigue. In future studies, it is important to evaluate the LSI rates at certain fatigue levels, particularly in athletes with an ACLR history, so that they can participate in physical activity with maximum effort. As the main hypothesis of our research was that the MAI technique would produce similar post-operative results as the conventional ST/G technique, the above-mentioned limitations were ignored. In addition, the fact that similar LSI rates of the SLHTs applied in different directions in both the ST/G and MAI techniques were obtained in our study may result from the similar rehabilitation of the subjects by the same physiotherapist.

## 5. Conclusions

The findings of our study revealed similar LSI rates of the SLHTs applied both traditionally and in different directions in both the ST/G and MAI techniques, which indicated that the MAI technique can be an ACLR technique which can be functionally used in athletes. However, in order to compare the MAI technique with other techniques, it is important to evaluate factors such as isokinetic strength tests, electromyographic analyses, and some performance components in addition to the SLHT in patients to whom different ACLR techniques have been applied.

## Figures and Tables

**Figure 1 medicina-58-00435-f001:**
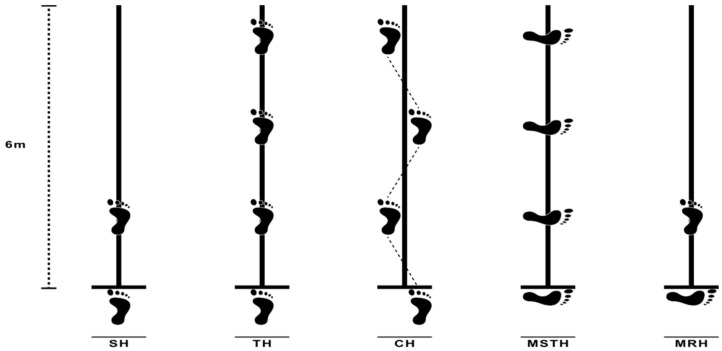
Graphical illustration of the single-leg hop tests.

**Figure 2 medicina-58-00435-f002:**
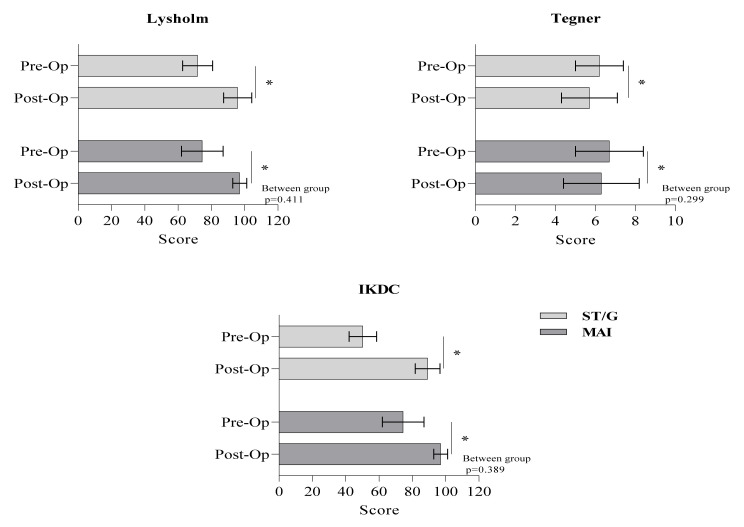
Comparison of pre-operative and post-operative levels of Lysholm, Tegner, and IKDC scores between techniques. * *p* < 0.05.

**Figure 3 medicina-58-00435-f003:**
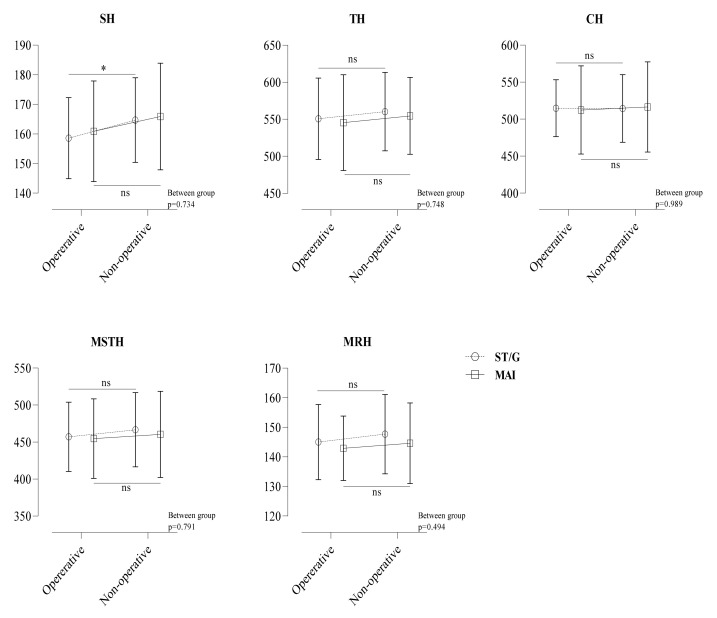
Differences in the hop test between operative and non-operative leg. * *p* < 0.05, ns = non-significant operative vs. non-operative.

**Table 1 medicina-58-00435-t001:** Descriptive statistics.

	ST/G (23)	MAI (16)	Total (39)
Age (years)	24.7 ± 5.4	27.6 ± 7.9	25.8 ± 6.6
Weight (kg)	78.3 ± 9	81.5 ± 14.1	79.6 ± 11.3
Height (cm)	179 ± 4.7	177.6 ± 5	178.4 ± 4.8
BMI (kg/m^2^)	24.4 ± 2.5	25.8 ± 3.8	25 ± 3.1
RTS (months)	6.7 ± 0.9	6.8 ± 1.1	6.7 ± 1

Data are presented as mean ± SD. No difference was observed between techniques (all comparisons had *p*-values greater than 0.05). MAI technique: modified all-inside technique; ST/G technique: semitendinosus/gracilis technique; RTS: return to sport time in months.

**Table 2 medicina-58-00435-t002:** Differences in the LSI between techniques.

	ST/G (23)	MAI (16)	*t*	*p*-Values
SH	96.6 ± 8	97 ± 7.3	0.147	0.884
TH	98.5 ± 7.1	98.3 ± 6.1	−0.079	0.938
CH	100.4 ± 6.7	99.6 ± 8.4	−0.035	0.732
MSTH	98.3 ± 6.8	99.2 ± 7.6	0.405	0.688
MRH	98.4 ± 6.5	99.2 ± 7.7	0.371	0.712
*p*-values	0.314	0.786		

Data are presented as mean ± SD. MAI technique: modified all-inside technique; ST/G technique: semitendinosus/gracilis technique.

## Data Availability

The datasets used and/or analyzed during the current study are available from the corresponding author on reasonable request.

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
