# Peer review of "Post-Operative Results of ACL Reconstruction Techniques on Single-Leg Hop Tests in Athletes: Hamstring Autograft vs. Hamstring Grafts Fixed Using Adjustable Cortical Suspension in Both the Femur and Tibia"

_medicina, 2022, doi:10.3390/medicina58030435_

Round 1
Reviewer 1 Report
Dear Authors, the article is very interesting. As regard the discussion i suggest to improve the point regarding the return to sport activity and performance and the second point regarding the evaluation of performance.
in fact it is important to underline that according to literature the return to sport is not associate to recovery of hight performance as before surgery. you could cite the following article:
“Returning to sport after anterior cruciate ligament reconstruction in amateur sports men: A retrospective study”Muscles, Ligaments and Tendons Journal Open AccessVolume 6, Issue 4, Pages 486 - 491October-December 2016Notarnicola A. et al
Author Response
Dear reviewer,
First of all, we would like to thank you for your positive comments about our research.
We have added the relevant parts of the article you recommended to the discussion section about the recommendations you have given about RTS. Thank you very much for your contribution to our article.

Reviewer 2 Report
Dear authors,
Your manuscript has a good idea behind it.
The presentation of the current problem is correct.
The description of materials and methods is done thoroughly. Maybe a picture describing the analyzed fixation methods would be appropriate. Another problem I have is with your group composition. Baseline characteristics of your patients from the 2 groups can vary a lot, as they do not seem matched in no way. Also, the low number pf patients decreases the statistical significance of your study. Since the study is retrospective, maybe try to gather a few more patients.
Results are presented in a nice manner with good statistical analysis.
I would concentrate a bit on the discussions, as all the information is hard to follow. Maybe consider organizing it in more paragraphs.
Author Response
Dear reviewer, first of all, we are grateful to you as authors for liking the idea and presentation of our research.
Maybe a picture describing the analyzed fixation methods would be appropriate.
Response: There are no fixation photos in the order that we can add to our research, but we will definitely follow your advice in our other researches about the current fixation method. Thank you very much for your opinion.
Another problem I have is with your group composition. Baseline characteristics of your patients from the 2 groups can vary a lot, as they do not seem matched in no way. Also, the low number pf patients decreases the statistical significance of your study. Since the study is retrospective, maybe try to gather a few more patients.
Response: You are quite right that there are small differences in the descriptive data of the subjects, but when we evaluated it statistically, we did not detect any significant difference. As you can imagine, the descriptive data of the patients we have reconstructed may differ.
Response: In addition, the fact that the number of our patients is not very large is one of the limitations of our research, as you said. However, as you can imagine, not all patients may come for follow-up after reconstruction or may not manage the rehabilitation process well. Therefore, we conducted the research on subjects who completed the whole process. In our further research, we will definitely design a study in which we can get more valid results by increasing the numbers. Thank you very much for your advice.
Results are presented in a nice manner with good statistical analysis.
Thank you so much
I would concentrate a bit on the discussions, as all the information is hard to follow. Maybe consider organizing it in more paragraphs.
Response: As you said, we have divided the discussion section into different paragraphs that are more interconnected. Thank you very much for your advice to make the discussion more readable.
Dear reviewer, thank you very much for your contribution to our article.
